# Clinical Outcomes of Patients with Metastatic Breast Cancer Treated with Hypo-Fractionated Liver Radiotherapy

**DOI:** 10.3390/cancers15102839

**Published:** 2023-05-19

**Authors:** Melinda Mushonga, Joelle Helou, Jessica Weiss, Laura A. Dawson, Rebecca K. S. Wong, Ali Hosni, John Kim, James Brierley, C. Anne Koch, Khalid Alrabiah, Patricia Lindsay, Teo Stanescu, Aisling Barry

**Affiliations:** 1Radiation Medicine Program, Princess Margaret Cancer Centre, Toronto, ON M5G 2M9, Canada; melinda.mushonga@kingstonhsc.ca (M.M.);; 2Department of Radiation Oncology, University of Toronto, Toronto, ON M5T 1P5, Canada; 3Department of Oncology, Division of Radiation Oncology, Western University, London, ON N6A 5W9, Canada; 4Department of Biostatistics, Princess Margaret Cancer Centre, University Health Network, Toronto, ON M5G 2M9, Canada; 5Department of Medical Biophysics, University of Toronto, Toronto, ON M5G 1L7, Canada; 6Cancer Research @UCC, University College Cork, T12 R229 Cork, Ireland

**Keywords:** liver metastasis, breast cancer, ablative radiotherapy, SBRT, SABR, metastases

## Abstract

**Simple Summary:**

Local ablative liver radiotherapy is increasingly being used in the setting of metastatic disease, primarily to prevent local disease progression and potentially to improve patient survival. There is a paucity of data specifically reporting the role of liver-directed ablative radiotherapy in metastatic breast cancer. The aim of this retrospective review was to report a single institutional experience in the use of hypo-fractionated liver radiotherapy, patient and treatment descriptors and treatment and disease outcomes. The study describes an excellent 1-year local control rate (100%), with an acceptable acute side-effect profile. Size of liver metastases was predictive of survival, with class of metastatic disease predictive for disease progression. Further prospective studies are required to assess the impact of metastatic classification on the indication of local ablative therapies, sequencing and outcomes post treatment in patients with metastatic breast cancer.

**Abstract:**

Purpose: To retrospectively review the clinical outcomes of patients with metastatic breast cancer (MBCa) following liver directed ablative intent radiotherapy (RT). Methods: Demographics, disease and treatment characteristics of patients with MBCa who received liver metastasis (LM) directed ablative RT between 2004–2020 were analysed. The primary outcome was local control (LC), secondary outcomes included overall survival (OS) and progression-free survival (PFS) analyzed by univariate (UVA) and multi-variable analysis (MVA). Results: Thirty MBCa patients with 50 LM treated with 5–10 fraction RT were identified. Median follow-up was 14.6 (range 0.9–156.2) months. Class of metastatic disease was described as induced (12 patients, 40%), repeat (15 patients, 50%) and de novo (three patients, 10%). Median size of treated LM was 3.1 cm (range 1–8.8 cm) and median biologically effective dose delivered was 122 (Q1–Q3; 98–174) Gy_3_. One-year LC rate was 100%. One year and two-year survival was 89% and 63%, respectively, with size of treated LM predictive of OS (HR 1.35, *p* = 0.023) on UVA. Patients with induced OMD had a significantly higher rate of progression (HR 4.77, *p* = 0.01) on UVA, trending to significance on MVA (HR 3.23, *p* = 0.051). Conclusions: Hypo-fractionated ablative liver RT in patients with MBCa provides safe, tolerable treatment with excellent LC.

## 1. Introduction

The liver is the third most prevalent site of metastatic spread in breast cancer, preceded by lung and bone [1]. Studies have shown such patients have poorer outcomes with a median survival of 2 to 3 years [2] and 5-year overall survival (OS) rates of only 8.5% [1].

Advancements in cross-sectional and functional imaging have allowed for improved identification of patients with limited metastatic disease. The literature describing outcomes following local treatment of limited metastatic lesions reports mixed clinical outcomes [3,4,5]. Some studies suggest breast cancer patients with oligo-metastatic disease benefit more from the use of ablative radiotherapy in comparison with other primary cancers [6,7], whilst two recent breast-specific studies suggest the contrary [5,8]. The European Society of Radiation Oncology (ESTRO)/European Organization of Research and Treatment of Cancer (EORTC) metastatic classification system has defined metastatic groups which will likely aid in streamlining clinical indications for treatment to better define the role of ablative RT in the metastatic setting [9,10].

The objective of this retrospective cohort study is to describe the clinical outcomes of patients with metastatic breast cancer (MBCa) who received hypo-fractionated liver radiotherapy (RT).

## 2. Methods

Breast cancer patients with liver metastases who received liver hypo-fractionated RT up to 10 fractions, between 2004 and 2020, were identified as part of a research ethics board approved retrospective study. Metastatic disease was classified as per the ESTRO/EORTC classification system [9]. The indication of treatment was classified as oligo-metastatic (OM; patients with limited metastatic disease with all metastases receiving ablative therapy) or oligo-progressive disease (OP; patients with widespread metastatic disease in the setting of limited progressing lesions). Liver radiotherapy planning has been discussed in previous publications [11,12]. Briefly, patients were simulated in a supine position with arms up on a chest board. Fluoroscopy was performed to assess for suitable motion management. Motion management options included active breathing control (ABC) in the exhale breath hold position or free-breathing with or without abdominal compression using an MR compatible belt. Patients underwent a multiphasic contrast-enhanced computed tomography (CT) scan of the liver, which included a four-dimensional scan with 0% (inhale) and 60% (exhale) phases if the patient was in free-breathing. Over time, a planning magnetic resonance imaging (MRI) with T1 and T2 sequences in arterial, venous and delayed venous phases were additionally used if a patient had no contraindications (i.e., pacemaker). Diagnostic scans (CT and MRI) were imported and coregistered to the simulation scan on the treatment planning system prior to contouring.

Gross tumour volume was contoured on the planning CT venous phase and if free-breathing in the 0% and 60% breathing phases also, with a 0 mm margin added to form the clinical target volume. In patients who underwent free-breathing simulation or used abdominal compression, an internal target volume was created by merging CTV volumes of all breathing phases. A planning target volume was created by adding a 5 mm isotropic expansion of the ITV (or clinical target volume if no ITV used). Organs at risk were contoured in the exhale phase dataset of the CT simulation scan.

Earlier plans were generated in Pinnacle (Elekta, Stockholm, Sweden), which was subsequently replaced by RayStation Laboratories treatment planning system. Flattening filter-free volumetric modulated arc therapy and intensity modulated radiation therapy were used for treatment delivery. At time of treatment delivery, a bone match was performed followed by a liver mask match. Patients were seen in the radiation oncologists review clinic once weekly while receiving RT. Follow-up imaging was completed as per institutional guidelines at 8–12 weeks post the final day of RT treatment then every 3–6 months.

Data collected from the electronic patient records and radiotherapy information system (e.g., Mosaiq—Elekta, Stockholm, Sweden). The following parameters were recorded: patient demographics, pathology, details on pre and post-RT systemic therapy, RT treatment details i.e., number of treated lesions, biologically effective dose (BED) of prescribed dose and number of fractions. When there were multiple gross target volumes, radiotherapy treatment details to the largest lesion were reported. An α/β ratio of 3 for local control, generated and informed by the START Pilot, START A, FAST and FAST FORWARD trials, was used for the biological equivalent dose (BED) calculation [13,14,15]. Information on acute toxicities was collected and toxicity scoring was described as per the National Cancer Institute-Common Terminology Criteria for Adverse Events version 4 (CTCAE) [16].

## 3. Outcomes and Statistical Analysis

The primary outcome was local control (LC) of the treated metastatic liver lesion measured from the pre-RT diagnostic CT. Local failure was defined as enlargement of the treated lesion measured on at least two consecutive computed tomography (CT) scans or MRI post-radiotherapy within the planning target volume. Secondary outcomes were progression-free survival (PFS)—defined as time to progression outside the treated field (i.e., in the liver but outside of the planning target volume, regional nodes or distant sites), OS and time to second-line systemic therapy post-liver hypo-fractionated RT.

Overall survival was calculated from completion of RT to the date of death or censored at last follow-up, PFS was calculated from treatment completion to the date of progression on imaging, date of death or last follow-up. For both time points, the hazard ratios (HR) and 95% confidence intervals (95% CI) were calculated using Cox proportional Hazards model. Time to second-line systemic therapy post-RT was measured as the time to initiation of a second-line systemic therapy from last day of RT calculated using the competing analysis method and death considered a competing risk. Logistic regression analysis was performed to assess for factors associated with OS, PFS and time to change of systemic therapy. Factors included in the univariate analysis (UVA) were, molecular subtype, treatment intent (OM/OP), EORTC classification (repeat; induced; de novo), lines of systemic therapy pre ablative liver RT, size of largest metastatic liver lesion and BED_3_ (<100 Gy; >100 Gy). Factors included in the multivariable analysis (MVA) of overall survival were treatment intent and lines of systemic therapy pre ablative liver RT based on clinical judgment and the commonly used rule of thumb of one variable per 10 patients. Additional variables were included if they were significant on UVA with a *p* value of <0.05. Statistical analysis was performed using the software package R v 3.4.2.

## 4. Results

Between June 2004 and October 2020, 30 patients with 50 liver metastases were treated with hypo-fractionated RT. The median age was 55.6 (range 32.1–79.3) years, with 21 (70%) patients having metastatic liver-only metastases at time of RT. Computed tomography was used to stage 93% of patients, with only two patients undergoing MRI with primovist contrast. The most common molecular subtype was Estrogen receptor (ER) positive (+ve)/Her 2-receptor negative (−ve) (17, 57%). As per the EORTC classification system, 12 (40%) patients had induced, 15 (50%) repeat and three (10%) de novo (one synchronous, two metachronous) oligo-metastatic disease (OMD). Oligo-progression (OP) of the liver metastases was the indication for treatment in 22 (73%) patients. Median diameter of the treated liver metastases was 3.15 (interquartile range {IQR} of 2.4–5.1) cm, median number of treated metastases was one (range 1–5) and median prescribed BED delivered was 122 (IQR of 97.9–174.3) Gy_3_ in 5–10 fractions. The most common number of fractions was five (14 patients), with 13 patients treated in six fractions. Only three patients (10%) received a ten fraction RT regimen (median BED_3_ 112.5 Gy) (Table 1).

### 4.1. Outcomes Post-Hypo Fractionated Radiotherapy Treatment

Median follow-up for the entire cohort was 14.6 months (range 0.9–156.2 months). The LC of treated metastatic lesions at 1 year was 100%. Only one local failure (patient classified as induced OMD receiving treatment for OP) was identified at 40 months post-RT in a patient who had ER/PR negative, HER2 positive breast cancer. This patient received a BED_3_ of 93 Gy in 10 fractions and was still alive at the time of last follow-up (13-years post-RT).

One year and two-year survival was 89% and 63% respectively. Larger size of treated liver metastasis (>3 cm) was the only variable predictive of worse survival on UVA; however, on MVA, it was not significant (*p* = 0.066) (Table 2).

Half of patients developed disease progression to other regional and distant sites outside of the treated liver metastases, with a median time to progression post-RT of 4.8 months (range 0.8–1145 months). The majority were patients with induced OMD (11 patients; 73%), followed by repeat (three patients; 20%) and de novo OMD (one patient; 7%). Oligo-progression was the most common indication for liver RT in patients who subsequently developed further disease progression (80%). The most common sites of distant failure were lung 33% (five of 15) and bone 27% (four of 15). Other sites included brain, lymph nodes and omental metastases.

Classification of metastases and indication of treatment were the only variables significantly associated with disease progression on UVA. Patients with induced OMD had a statistically significant higher rate of progression (HR 4.77, *p* = 0.01) on UVA compared to others, which trended to significance on MVA (HR 3.23, *p* = 0.051). Disease progression outside of the treated liver was more likely in patients where the indication of treatment was oligo-progression (HR 3.72, *p* = 0.044) (Table 2 and Table 3).

### 4.2. Pre-Hypo Fractionated RT and Post-Hypo Fractionated RT Systemic Treatment

All patients, but one, received some form of systemic therapy from initial diagnosis to receipt of ablative liver RT (Table 1). The median lines of pre-ablative RT systemic therapy received by all patients were two (range 0–6). Seventeen patients (56%) were switched to next line systemic therapy post-liver RT, ten of whom had regional or distant disease progression and seven without indication of progression. The median time to second-line ST after RT was 9.9 (range 1–114) months. The median time for patients initiated on second-line systemic therapy at disease progression having continued pre-RT systemic therapy was 5.4 (range 2–114) months in comparison with 19.1 (4.4–26.4) months for patients initiated on another ST at time of RT. No variable on UVA was predictive of switching to second-line systemic therapy post-RT (Table 4).

### 4.3. Acute and Late Toxicities

Acute grade 1 and 2 toxicities were reported in 20/30 (66%), with the most common acute toxicities of nausea (68%) and fatigue (61%) reported. One patient developed an acute CTCAE grade 3 RT-related gastric ulcer within 3 months of treatment confirmed by endoscopy after presenting with hematemesis. The patient had two liver metastases treated. One of the lesions was in segment three of the liver adjacent to the stomach and was prescribed 30 Gy in five fractions every other day. The second smaller lesion in segment six was prescribed 40 Gy in fiver fractions. An abdominal compression system was used for motion management including image guidance for each treatment session. Prophylactic proton pump inhibitors (PPIs) were prescribed during and after liver RT for this patient as the metastasis was close to the stomach. The maximum dose to the stomach was 31 Gy and a corresponding mean of 12 Gy. Gastritis was managed by medication alone with PPI and the patient recovered clinically by the next follow-up. No late CTCAE grade 2 or higher radiotherapy-related toxicities were recorded.

## 5. Discussion

Hypo-fractionated liver RT, regardless of metastatic classification, resulted in excellent LC at 1-year of 100% of treated metastatic liver lesions. These outcomes are consistent with current literature describing patients with metastatic breast cancer treated with liver SBRT, reporting 1 and 2-year LC rates of 88–100% and 75–88%, respectively [17,18,19].

Similarly, one and two-year overall survival rates of 89% and 63% described in this study were comparable to that reported in the literature [17,18,19]. The only significant variable associated with worsened survival in this study was the size of the treated liver metastases (>3 cm). Larger metastases have previously been associated with poorer LC in patients with metastatic breast cancer receiving curative intent SBRT [20], attributed to an increased risk of micro-metastatic disease in patients with larger disease burden [21], therefore increased risk of distant failure and poor survival.

About half of patients described developed early disease progression, with a median PFS of 4.8 months though there was no difference by metastatic class (OM vs. OP). There is a paucity of clinical data describing breast cancer patient outcomes as per ESTRO/EORTC classification system. Induced oligo progression has been shown to be associated with worse survival in non-breast cancer patients. [22]. Tan et al. retrospectively described a cohort of 120 patients with metastatic breast cancer treated with SBRT to various anatomical sites (24 patients had liver SBRT), noting a difference in OS and PFS in favour of patients with OMD compared to OP disease [23]. These findings demonstrating a difference based on metastatic classification suggest classification of metastatic disease should be routinely considered in prospective studies assessing the role of ablative therapies in the management of metastatic disease to guide suitable patient selection.

In patients with induced OMD, we noted that in our study these patients had a higher rate of distant progression compared to patients with repeat or de novo OMD. This is unsurprising as such patients are likely at higher risk of relapse given a history of prior poly-metastatic disease, compared to those without, and are more likely to have had multiple lines of systemic therapy which may result in a greater propensity for future clonal resistance. The recent phase IIR NRG BR002 trial [5] included non-classified oligo-metastatic breast cancer patients defined as patients with four or less extra-cranial metastatic lesions. Patients were required to have a controlled primary and had to be on standard of care ST for ≤12 months prior to enrolment. The study compared ablative therapy (93% SBRT, 7% surgery) to all lesions with ST versus ST alone. They reported a 2-year PFS of 46.8% [5] with no difference in median PFS in patients who received the combination arm compared to ST alone (19.5 vs. 23 months, *p* = 0.92). Acknowledging our study is retrospective, therefore, a different study design, the mixed findings highlight the need for trials with pre-defined subgroups to inform which cohort of breast cancer patients may benefit from ablative therapy in the setting of limited metastases and further inform sequencing with ST.

Oligo-progression was the most common treatment indication in this study (73%). This group of patients is commonly defined as having widespread metastatic disease with a limited number of metastases progressing on systemic therapy and are less well studied with unanswered questions on the right timing to initiate systemic therapy [24]. The CURB study [8], a prospective randomized breast cancer study that treated patients with ≤5 progressing metastatic lesions with standard-of-care ST with or without SBRT, found no difference in median PFS between patients who received SBRT (*n* = 24) and those who did not (*n* = 23) (18 vs. 19 weeks, *p* = 0.47). One reason may be that patients with OP disease are often heavily pretreated with systemic therapies and harbour radioresistant clones resulting in inferior LC post-ablative RT [25]. Patients with OP in our study received more lines of pre-RT systemic therapy compared to patients with OMD, but this did not appear to affect LC rates. A common clinical indication for ablative RT in patients with OP disease is to delay start of next line ST, especially in the setting of limited lines of ST. In this cohort, some patients were switched to another ST in the immediate period after RT and had a longer interval of time to switch to second-line ST post-RT compared to those switched at time of progression after continuing pre-RT systemic therapy.

Breast cancer is generally recognized as a radiosensitive disease; however, there is limited data on the impact of molecular subtypes on radiosensitivity. With the recently updated TNM staging accounting for molecular classification, it has not yet been established if radiotherapy should be altered to account for the potential variation in biology. Evidence supporting dose escalation in the setting of K-RAS mutant colorectal [26,27,28,29,30] cancer suggests intrinsic biology matters; however, neither molecular subtype or BED >100 Gy_3_ impacted primary or secondary outcomes in this retrospective review. It is poorly understood how different molecular subtypes respond to RT [31]. Yard et al. described breast cancer cell lines with elevated *ERBB2*, being associated with radiation resistance prior to Her 2 targeted systemic therapy introduction [32]. Subsequent studies have shown that trastuzumab, sensitises breast cancer cells to radiation by enhancing radiation induced apoptosis conferring radiosensitivity which ultimately translates to superior LC [33]. The small subset of Her 2 positive patients precluded a meaningful analysis to compare with findings from precedent studies described.

Radiation dose is an established predictor of local control [34], and in this study local control was excellent regardless of dose received. In a study by Klement et al. who investigated tumour control in patients with liver metastases receiving SBRT found that compared with other histology, breast cancer had the strongest relationship between LC and largest BED (using an alpha/beta 10) [25]. Conversely, Tan et al., found no association between BED (using an alpha/beta 3) or molecular subtype with local control in breast cancer patients receiving SBRT to extracranial metastases [23].

Toxicity rates described are comparable to those reported in literature, with no acute or late CTCAE grade 4 or 5 toxicities and only one patient developing a CTCAE grade 3 GI toxicity (gastric ulcer) [17]. On review the patient had two liver lesions were treated, dose volume constraints met standard criteria and appropriate premedication was prescribed as per institutional guidelines. However, this highlights the need to remain cognizant of potential treatment-related side effects in patients receiving liver SBRT. Adoption of advanced technology, such as daily adaptive planning with the MRI linear accelerator may be an appropriate measure for lesions in dose-limiting locations close to organs at risk.

Due to its retrospective nature, this study has limitations which include the presence of missing data, reporting bias, selection bias and confounders such as use of pre- and post-ablation systemic therapy and evolving systemic treatment options during the study period. Additionally, heterogeneity of molecular subtype, dose/fractionation and lack of specificity in systemic therapy data (actual sequencing of type of systemic therapy by molecular subtype and duration on a specific therapy) limits interpretation of findings in clinical practice but rather guides on areas of study to focus on. The treatment of the other distant sites in nine patients was not explored and when they were multiple targets, radiotherapy treatment details to the largest lesion were the only one reported. Additionally, over the past two decades (during which these patients were treated), there have been many changes to how patients are clinically managed, such as radiological staging (CT vs. MRI) and systemic therapies.

Several interesting questions have been brought forward regarding the evolution of the integration of ablative RT in the setting of metastatic breast cancer. It is apparent on literature review that there is a paucity of data describing patients with breast cancer according to the different classes of metastatic disease, indications of treatment, systemic therapies and molecular subtypes [35]. Such variables should guide future studies with the aim to better prognosticate patients [36] who could benefit from ablative radiotherapy and optimum sequencing with evolving modern systemic therapies [37], aiding clinician and patient decision making.

## 6. Conclusions

Hypo-fractionated ablative liver radiotherapy in patients with MBCa provides safe, tolerable treatment with excellent 1 year LC. Further studies are needed to identify patients, with MBCa according to metastatic class, who might benefit from ablative RT.

## Figures and Tables

**Table 1 cancers-15-02839-t001:** Patient, disease, and treatment characteristics.

Variable	*n* = 30
**Age**	
Median, years (range)	55.6 (32.1–79.3)
**Eastern Cooperative Oncology Group (ECOG) Performance status**	
0 or 1	26 (87)
2	4 (13)
**Histology**	23 (77)
Invasive ductal carcinoma	4 (13)
Invasive lobular carcinoma Others	3 (10)
**Molecular subtype**	
ER +ve/Her 2 −ve	17 (57)
ER +ve/Her 2 +ve	7 (23)
ER −ve/Her 2 +ve	2 (7)
ER −ve/Her 2 −ve	4 (13)
**Number of initial metastatic sites**	
1 (Liver only)	21 (70)
2	6 (20)
3	3 (10)
**Time from primary diagnosis to ablative radiotherapy to liver lesions**	
Median, months (Q1–Q3)	37.1 (18.5–73.3)
Number of treated liver metastases	
Median (range)	1 (1–5)
**Number of treated liver metastases (*n* = 50)**	
1	17
2	6
3	4
4	2
5	1
**Metastatic classification**	
Induced OM	12 (40%)
Repeat OM	15 (50%)
De novo OM	3 (10%)
**Lines of systemic therapy pre-ablative radiotherapy to liver lesions**	
Median (range)	2 (0–6)
**Type of systemic therapy pre-ablative radiotherapy**	
Endocrine therapy	24
Anti Her 2 therapy	8
Systemic Chemotherapy	23
Cyclin Dependent Kinase 4/6 inhibitor	3
mTor inhibitors	2
Other targeted treatments	1
**Initial diameter of largest metastatic lesion for treatment (cm)**	
Median (Q1–Q3)	3.15 (2.4–5.1)
**Treatment Indication**	
OP	22 (73)
OM	8 (27)
**Systemic therapy post-ablative radiotherapy to liver lesions**	
Yes	28 (93)
No	1
Missing	1
**Type of first-line systemic therapy post-ablative radiotherapy**	
Endocrine therapy	17

Acronyms—ER—Estrogen; +ve—positive; −ve—negative; OM—Oligometastases; OP—Oligoprogression; RT—Radiotherapy; BED—Biological equivalent dose.

**Table 2 cancers-15-02839-t002:** Univariate and Multivariable Analysis of Overall Survival.

Variable	Overall Survival					
	Univariate Analysis			Multivariable Analysis		
	HR	95% CI	*p*-Value	HR	95% CI	*p*-Value
Molecular subtype (ER +ve/Her 2 −ve Others)	1.03	0.33–3.22	0.96			
Treatment Intent OM OP	4.59	0.99–21.37	0.052	4.32	0.97–21.58	0.074
EORTC classification Repeat Induced De Novo	Reference 2.04 2.25	0.56–7.45 0.42–11.94	0.46 *			
Lines of systemic therapy pre ablative liver RT	1.36	0.93–1.98	0.11	0.92	0.58–1.45	0.72
Size of largest metastatic liver lesion	1.35	1.04–1.75	0.023	1.34	0.98–1.84	0.066
BED_3_ <100 >100	1.27	0.27–6.09	0.76			

Acronyms—ER—Estrogen; EORTC—European Organization for Research and Treatment of Cancer; OM—Oligometastases; OP—Oligoprogression; RT—Radiotherapy; BED—Biological equivalent dose; HR—Hazard ratio; CI—Confidence Interval. * Global *p*-value.

**Table 3 cancers-15-02839-t003:** Univariate and Multivariable Analysis of Progression-Free Survival.

Variable	Progression Free Survival					
	Univariate Analysis			Multivariable Analysis		
	HR	95% CI	*p*-Value	HR	95% CI	*p*-Value
Molecular subtype (ER +ve/Her 2 −ve Others)	0.91	0.35–2.36	0.84			
Treatment Intent OM OP	3.72	1.04–13.3	0.044	3.27	0.67–16.01	0.14
EORTC classification Repeat Induced De Novo	reference 4.77 3.47	1.44–15.80 0.61–19.88	0.022 *	reference 3.23 6.75	0.95–10.96 1–45.38	0.051 *
Lines of systemic therapy pre -ablative liver RT	1.29	0.95–1.75	0.10	1.14	0.76–1.69	0.53
Size of largest metastatic liver lesion	1.14	0.93–1.39	0.21			
BED_3_ <100 >100	0.39	0.14–1.06	0.066			

Acronyms—ER—Estrogen; EORTC-European Organization for Research and Treatment of Cancer; OM—Oligometastases; OP—Oligoprogression; RT—Radiotherapy; BED—Biological equivalent dose; HR—Hazard ratio; CI—Confidence Interval. * Global *p*-value.

**Table 4 cancers-15-02839-t004:** Univariate Analysis of Time to Next Line Systemic Therapy.

Variable	Time to Next Line Systemic Therapy		
	Univariate Analysis		
	HR	95% CI	*p*-Value
Molecular subtype ER +ve/Her 2 −ve Others	0.85	0.32–2.23	0.74
EORTC classification Repeat Induced De Novo	Reference 1.25 1.23	0.45–3.53 0.46–3.26	0.88 *
Lines of systemic therapy pre ablative liver RT	1.27	0.99–1.64	0.057
Size of largest metastatic liver lesion	1.17	0.95–1.44	0.15
BED_3_ <100 >100	0.47	0.12–1.88	0.29
Treatment intent OM OP	Reference 1.27	0.51–3.17	0.6

Acronyms—ER—Estrogen; EORTC-European Organization for Research and Treatment of Cancer; OM—Oligometastases; OP—Oligoprogression; RT—Radiotherapy; BED—Biological equivalent dose; HR—Hazard ratio; CI—Confidence Interval. * Global *p*-value.

## Data Availability

Data is available on request from the reporting institution.

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
