# Peer review of "Clinical Outcomes of Patients with Metastatic Breast Cancer Treated with Hypo-Fractionated Liver Radiotherapy"

_cancers, 2023, doi:10.3390/cancers15102839_

Round 1

Reviewer 1 Report

Thank you for the opportunity to review this manuscript.

Data on radiotherapy of liver metastases is important, even more after the contradictive situation after results of the "NRG study".

Your rate of adverse events seems to be comparatively high (same accounts for Onal et al. 2018 who described 2 patients with Grade 3 events out of 22 patients). Thus, I would suggest to comment on this circumstance in the discussion. 

This is an important topic, since if the rate of adverse events is high, then the indication of liver SBRT in OMD/OPD Breast cancer is even more to be indicated very cautiously and under special circumstances. Finally, it would be advisable to briefly comment on novel treatment approaches, e.g. MR-guided radiotherapy, which is (potentially) less toxic (but nonetheless very ressource intensive)

Author Response

Dear Reviewer 1,

Many thanks for taking the time to review our manuscript. Please find our responses below

Data on radiotherapy of liver metastases is important, even more after the contradictive situation after results of the "NRG study".

  • Your rate of adverse events seems to be comparatively high (same accounts for Onal et al. 2018 who described 2 patients with Grade 3 events out of 22 patients). Thus, I would suggest to comment on this circumstance in the discussion.
  • This is an important topic, since if the rate of adverse events is high, then the indication of liver SBRT in OMD/OPD Breast cancer is even more to be indicated very cautiously and under special circumstances. Finally, it would be advisable to briefly comment on novel treatment approaches, e.g. MR-guided radiotherapy, which is (potentially) less toxic (but nonetheless very resource intensive)

We agree and the following has been added - Discussion line 281 - 289 -

'Toxicity rates described are comparable to those reported in literature, with no acute or late CTCAE grade 4 or 5 toxicities and only one patient developing a CTCAE grade 3 GI toxicity (gastric ulcer). 17On review the patient had 2 liver lesions were treated, dose volume constraints met standard criteria and appropriate premedication was prescribed as per institutional guidelines. This, however, highlights the need to remain cognizant of potential treatment related side effects in patients receiving liver SBRT. Adoption of MRI linac guided treatment where feasible and resource permitting may be an appropriate measure for lesions in critical locations.'

Reviewer 2 Report

  1. Size of largest lesion univariate vs multivariate analysis suggests that the lesion size is only predictive of overall survival due to its association with other factors in multivariate analysis. This discrepancy in results was not adequately explored. Which other factors included in the multivariate analysis could be more relevant to overall survival? (LINE 209)
  2. While the authors state the paucity of data in regards to using EORTC classification as criterion, the study fails to reach any concrete conclusion due to lack of power. To detect differences in outcomes, any observable trending towards significance (p=00051) in multivariate analysis was insufficient.  The authors could consider expanding their years of data collection from hospital records to increase patient sample volume to increase power.
  3. Line 226, the authors state: "In patients with induced OMD, we noted that in our study these patients had a higher 226 rate of distant progression compared to patients with repeat or de novo OMD". Please explain why the authors think they saw this trend.
  4. Line 228: The relation between phase IIR trial and EORTC classification is unclear. The authors need to state why, if at all, there would be any difference in this study’s definition of OMD classification to EORTC’s classification. Would there a functional difference in outcomes? If so, please state it explicitly. Otherwise, omit mentioning the differences in classification. Please consider rephrasing this paragraph by discussing the study first before interjecting with EORTC classification and drawing comparisons.
  5. Since all the multivariate analyses yielded non-significant results, please state how the authors could draw any conclusions about long-term safety with this data. 

Author Response

Dear Reviewer 2,

Many thanks for taking the time to review our manuscript. Please find below our replies to your comments - 

  • Size of largest lesion univariate vs multivariate analysis suggests that the lesion size is only predictive of overall survival due to its association with other factors in multivariate analysis. This discrepancy in results was not adequately explored. Which other factors included in the multivariate analysis could be more relevant to overall survival? (LINE 209)

Size of liver metastases was significant on UVA (p=0.023) for overall survival only. Following the general rule of thumb of 1 variable per 10 patients, 2 additional variables were added to the MVA. Treatment intent (p=0.052) was borderline significant on UVA and so included, while lines of systemic therapy was included due to potential for clinical significance.

On analysis of progression free survival, 2 variables were SS on UVA - treatment intent and classification, which left room for one additional variable to be included in MVA. Lines of systemic therapy was deemed the most clinically significant and therefore incorporated into the MVA. 

No variables was significant for time to change in next line of systemic therapy on UVA. 

Unfortunately number of patients included is the limiting factor, we could of course continue to explore but we would be at risk of biasing results.

The following as been added - 

Methods, paragraph 2, line 121

'Factors included...of overall survival.....and the commonly used rule of thumb of one variable per 10 patients.'

  • While the authors state the paucity of data in regards to using EORTC classification as criterion, the study fails to reach any concrete conclusion due to lack of power. To detect differences in outcomes, any observable trending towards significance (p=00051) in multivariate analysis was insufficient.  The authors could consider expanding their years of data collection from hospital records to increase patient sample volume to increase power.

We agree and were surprised at the small numbers treated, however, this study was from 2004- 2020 and analysis completed in 2021. Additional years would be 2021-2022 only, therefore any additional patients are likely minimal to detect any further meaningful changes from the original analysis, especially as this is a very selective patient cohort- breast cancer only with liver metastases for HFRT

  • Line 226, the authors state: "In patients with induced OMD, we noted that in our study these patients had a higher 226 rate of distant progression compared to patients with repeat or de novo OMD". Please explain why the authors think they saw this trend.

The following has been added - Discussion, lines 230 -234

'This is unsurprising as such patients are likely at higher risk of relapse given a history of prior poly-metastatic disease, compared to those without, and are more likely to have had multiple lines of systemic therapy which may result in a greater propensity for future clonal resistance. '

  • Line 228: The relation between phase IIR trial and EORTC classification is unclear. The authors need to state why, if at all, there would be any difference in this study’s definition of OMD classification to EORTC’s classification. Would there a functional difference in outcomes? If so, please state it explicitly. Otherwise, omit mentioning the differences in classification. Please consider rephrasing this paragraph by discussing the study first before interjecting with EORTC classification and drawing comparisons.

Agree, the line has been removed.

  • Since all the multivariate analyses yielded non-significant results, please state how the authors could draw any conclusions about long-term safety with this data. 

The conclusion on long term safety was derived from the rates of toxicity described, which is an additional secondary outcome. These were not included in UVA or MVA for OS, PFS or time to next line systemic therapy. Given there was only 1 grade 3 toxicity we did not think by adding such a variable to the UVA/MVA analysis would add to deriving a potential correlation to OS/PFS/Time to next line systemic therapy. Also our toxicity results are consistent with published literature so stand by that we believe SBRT to be safe and effective in the local management of this disease. This does not however answer the question of the impact of using such a therapy on survival outcomes for these patients, but adds to the literature. It challenges us clinicians to better design future prospective trials which seek to address clinical practice management questions.

Reviewer 3 Report

In this study, the authors retrospectively analyse the effect of liver RT on OS, OFS, and switch to the next chemotherapy in breast cancer patients with liver metastasis. One major drawback that compromises the significance of the study is the number of cases included in the study. Only 30 cases have been identified from the clinical information obtained during 2004-2020. It is a pity.  

Comments

1.       There are too many abbreviations used in this manuscript. The authors might consider including a list showing all the abbreviations. Therefore, the readers will be easier to identify the items.

2.       In Table 1, there is an item called “Performance status”. What is it? Regarding molecular subtypes, why do the authors omit PR status? Also, the authors do not mention the metastatic liver lesion sizes, but I can find some information in the abstract.

3.       In line 149, the authors mention, "One year and two-year survival was 89% and 63% respectively”. However, I cannot find the corresponding analysis in the result resections.

4.       In Table 2, there is an item called “reference” in EORTC classification? What is it? Also, the number of patients used for the analysis should be reported.

5.       In Table 4, the three items “ Molecular subtype ER+ve HER-ve Others” are on the same line. The author should revise it. And How about the multivariate analysis?

6.       In section 4.3, do the authors check with the liver function pre-and post-RT? In addition, do the authors identify any disease related to liver damage?

7.       Based on Table 1, most patients received endocrine therapy or systematic chemotherapy. Given that the patients had the same subtype, do the authors determine if a particular treatment would be associated with a better response to RT? 

Author Response

Dear Reviewer 3,

Many thanks for taking the time to review our manuscript. Please find below our responses to your comments and queries -

  • There are too many abbreviations used in this manuscript. The authors might consider including a list showing all the abbreviations. Therefore, the readers will be easier to identify the items.

Many thanks for this, we have reviewed the manuscript and reduced the number of abbreviations.

  • In Table 1, there is an item called “Performance status”. What is it? Regarding molecular subtypes, why do the authors omit PR status? Also, the authors do not mention the metastatic liver lesion sizes, but I can find some information in the abstract.

'Performance status' is the Eastern Cooperative Oncology Group Performance Status which has been clarified in the table.

PR (progesterone) status was not included as many patients did not have the result recorded in their pathology or their clinical notes.

The size of liver lesions is included in Table 1, just before Treatment indication.

  • In line 149, the authors mention, "One year and two-year survival was 89% and 63% respectively”. However, I cannot find the corresponding analysis in the result resections.

This is present in the results section, paragraph 4.1 Outcomes post hypo-fractionated RT, line 154.

  • In Table 2, there is an item called “reference” in EORTC classification? What is it? Also, the number of patients used for the analysis should be reported.

'Reference' is where there are 2 or more categories used in the analysis. The reference category is the one used for comparison. Number of patients is 30 and proportions used for each analysis is indicated in Table 1.

  • In Table 4, the three items “ Molecular subtype ER+ve HER-ve Others” are on the same line. The author should revise it. And How about the multivariate analysis?

Noted and adjusted.

No MVA was performed as there were no statistically significant variables and we did not think it was methodologically appropriate to try and 'find' more outcomes given limitations of patient numbers.

  • In section 4.3, do the authors check with the liver function pre-and post-RT? In addition, do the authors identify any disease related to liver damage?

We agree this would be important to add. Unfortunately we were limited due to the retrospective nature of this study in reviewing liver function tests. The clinical notes for each of these patients did not describe any pertinent liver related disease such as cirrhosis or hepatitis.

  • Based on Table 1, most patients received endocrine therapy or systematic chemotherapy. Given that the patients had the same subtype, do the authors determine if a particular treatment would be associated with a better response to RT?

We initially hope to be able to describe if outcome was related to hormonal status. Just over half of patients had ER+/HER2-ve disease, but when we compared this group to 'others' we did not identify any significant association with the outcomes chosen. Nor did we see that the use of pre-systemic therapy was significant.

We agree this question is of great importance in this population and therefore highlights the need to incorporate such variables into prospective trials exploring optimum sequencing of treatment modalities by molecular subtype in the metastatic setting.

Round 2

Reviewer 3 Report

The authors have addressed all my concerns.